# Zinc Oxide-Loaded Cellulose-Based Carbon Gas Sensor for Selective Detection of Ammonia

**DOI:** 10.3390/nano13243151

**Published:** 2023-12-15

**Authors:** Hao Xu, Zhu-Xiang Gong, Li-Zhu Huo, Chao-Fei Guo, Xue-Juan Yang, Yu-Xuan Wang, Xi-Ping Luo

**Affiliations:** Zhejiang Provincial Key Laboratory of Chemical Utilization of Forestry Biomass, College of Chemistry and Materials Engineering, Zhejiang A&F University, Hangzhou 311300, China; jasonxu@stu.zafu.edu.cn (H.X.); 2022005@stu.zafu.edu.cn (Z.-X.G.); hlz@stu.zafu.edu.cn (L.-Z.H.); chaoguo@zafu.edu.cn (C.-F.G.); yangxj@zafu.edu.cn (X.-J.Y.)

**Keywords:** gas sensor, ammonia, zinc oxide, microcrystalline cellulose

## Abstract

Cellulose-based carbon (CBC) is widely known for its porous structure and high specific surface area and is liable to adsorb gas molecules and macromolecular pollutants. However, the application of CBC in gas sensing has been little studied. In this paper, a ZnO/CBC heterojunction was formed by means of simple co-precipitation and high-temperature carbonization. As a new ammonia sensor, the prepared ZnO/CBC sensor can detect ammonia that the previous pure ZnO ammonia sensor cannot at room temperature. It has a great gas sensing response, stability, and selectivity to an ammonia concentration of 200 ppm. This study provides a new idea for the design and synthesis of biomass carbon–metal oxide composites.

## 1. Introduction

Ammonia is a colorless, toxic, and corrosive substance with a strong odor and a choking effect. Excess ammonia can pose a huge threat to human health and cause environmental pollution. Different from other toxic gases, ammonia has a low boiling point of −33.5 °C, a low melting point of −77.75 °C, a low density of 0.771 g·L^−1^, a refractive index of 1.33, and a dipole moment of 1.42 d. These characteristics make ammonia an excellent gas sensor, and its sensors can be used in a variety of applications, such as environmental monitoring, agriculture, medical diagnostics, and dealing with industrial waste [1,2].

Recently, metal oxide semiconductors (MOSs), such as ZnO [3], MoO_3_ [4], TiO_2_ [5], WO_3_ [6], NiO [7], and V_2_O_5_ [8], etc., have been widely used in the field of gas sensing due to their excellent physical and chemical properties. A gas sensor based on a TiO_2_/Ti_3_C_2_T_x_ bilayer film was fabricated by Tai et al. which can operate at room temperature but is only 3.1% responsive to a concentration of 10 ppm ammonia [5]. Chou et al. fabricated a NiO thin-film ammonia sensor using a radio frequency sputtering process. This ammonia sensor showed a good sensing performance to 1000 ppm ammonia; however, this sensor needs to be at 250 °C to achieve the optimal gas sensing performance [7]. Zinc oxide is a metal oxide semiconductor material with a wide band gap (3.37 eV), high bonding strength, and large exciton binding energy (60 meV) at room temperature. Based on these properties, it is often used in gas sensors, chemical sensors, biosensors, cosmetics, energy storage, optical and electronic devices, and other products [9,10,11,12,13]. The existing zinc oxide gas sensors show great gas sensing performance for carbon dioxide, ammonia, and ethanol. A resistive gas sensor based on ZnO nanosheets was prepared by Srinivasulu et al. which has a large surface area and can quickly and highly detect carbon dioxide in the air [14]. The ZnO nanoflowers prepared by Yu et al. have great gas sensing properties for low concentrations of ammonia [3]. However, the original zinc oxide still has some shortcomings, such as poor gas sensitivity, slow response and recovery time, and poor selectivity. Therefore, many researchers have used methods such as doping with other elements or mixing zinc oxide with other metal oxides to prepare composite materials to improve the gas sensing performance [15,16,17,18,19].

An effective way to improve the gas sensing performance of the sensor is by introducing carbon material into the gas sensing material based on zinc oxide. Using activated carbon fiber as a template, Chao et al. synthesized ZnO/C nanoporous fibers using a hydrothermal method. The materials showed a great gas sensing performance for low concentrations of ethanol and acetone at the optimum operating temperature [20]. The heterojunction ZnO/hollow porous carbon microtubule (CMT) prepared by Sun et al. from simple carbonized buttonwood fluff fibers showed a great gas sensing response to trace ammonia molecules [21]. Hu et al. prepared composite materials by generating zinc oxide particles on nitrogen-doped carbon sheets, and the heterogeneous structure improved the gas sensing ability of the composite materials to ppb grade NO_2_ [22]. Cellulose, as a natural polymer material abundant on the earth, has a large number of hydroxyl functional groups, excellent biodegradability, and stable physical and chemical properties [23]. Compared with other carbon sources, cellulose is more available in nature, which can reduce the cost of producing carbon materials significantly. In recent years, there have been many reports on the application of cellulose-based carbon (CBC) as a carbon material in the field of supercapacitors and photocatalytic degradation by taking advantage of its excellent properties, such as its porous structure, huge specific surface area, and ability to adsorb gas molecules and macromolecular pollutants [24,25,26,27,28]. However, there is little research on carbonizing cellulose into biomass carbon materials into ZnO-based gas sensors.

In this work, zinc oxide was prepared using a simple chemical precipitation method, and then it was carbonized with microcrystalline cellulose at a high temperature, and ZnO/CBC was prepared for ammonia detection. Compared with the original ZnO-based ammonia sensor, due to the construction of the heterogeneous structure between ZnO and CBC, the composite has a great gas sensing response to a certain concentration of ammonia at room temperature and has a better ammonia selectivity. The use of cellulose as the precursor of biomass carbon materials combined with metal oxides to form composite materials provides a new reference in the field of gas detection.

## 2. Materials and Methods

### 2.1. Materials

Zinc acetate dihydrate (Zn(CH_3_COO)_2_·2H_2_O, 98%) was obtained from Anneji (Shanghai) Pharmaceutical Chemical Co., Ltd. (Shanghai, China). Ammonia liquor (NH_3_·H_2_O, 26–28%) was purchased from Sinopharm Chemical Reagent Co., Ltd. (Shanghai, China). Microcrystalline cellulose (50 μm) was obtained from Aladdin Reagent (Shanghai) Co., Ltd. (Shanghai, China). Deionized water was prepared in the laboratory.

### 2.2. Synthesis of ZnO/CBC

ZnO was prepared using a simple chemical deposition method. Typically, 0.25 g zinc acetate dihydrate and 0.5 g microcrystalline cellulose were added into 30 mL DI water with magnetic stirring for 15 min. Then, a certain amount of ammonia (to make the molar ratio of Zn^2+^ to OH^−^ 1:15) was added to the above solution, with magnetic stirring in an oil bath at 90 °C for 1 h. After the reaction, the obtained white solution was washed and centrifuged several times until the pH was neutral and then dried in a vacuum drying oven at 60 °C to obtain a white powder. The white powder was placed in a tube furnace at 600 °C and calcined by N_2_ for 2 h. The resulting product was marked as ZnO/CBC-60%. According to the above experimental method, ZnO/CBC-33%, ZnO/CBC-50%, ZnO/CBC-66%, ZnO/CBC-71%, and CBC were synthesized by changing the mass ratio of zinc acetate dihydrate and microcrystalline cellulose.

### 2.3. Characterization

The morphology and size of the samples were characterized with field-emission scanning electron microscopy (SEM, ZEISS Sigma 300, Germany) and transmission electron microscopy (TEM, FEI Tecnai F30, USA). The crystal structure of the sample was characterized with X-ray diffraction (XRD, Bruker D2 PHASER, Germany) of copper target radiation at ambient temperatures with 2θ values of 10 to 80 degrees and scanning rates of 5°/min. The chemical composition and elements of the samples were characterized with X-ray photoelectron spectroscopy (XPS, Thermo Scientific K-Alpha, USA). Raman spectra were recorded using a Raman spectrometer (Raman, Horiba LabRAM HR Evolution, Japan) with a laser wavelength of 532 nm. In situ DRIFT spectroscopy was performed using a VERTEX 80v (Bruker, Germany) infrared spectrometer. ZnO/CBC-60% was exposed to ammonia and pure nitrogen at room temperature. The Mott–Schottky curves of the samples were measured with an electrochemical workstation (Chinstruments, CHI660E model, China); 0.5 M sodium sulfate solution was configured, Ag/AgCl was the reference electrode, and the platinum plate was the counter electrode.

### 2.4. Fabrication of Gas Sensing Device

In order to measure the electrical and sensing performance, a certain amount of sample was placed on an agate mortar, and a small amount of deionized water was added to grind the sample to a sticky state. The grinding liquid was evenly coated on the interdigital electrode of the ceramic substrate and dried in a drying oven at 100 °C. Gas sensing measurements are carried out using the self-made device shown in Figure 1. The static liquid–gas distribution method was used to generate ammonia gas environments with different concentrations. The calculation formula is as follows [29,30]:(1)$$C=\frac{22.4\times \emptyset \times \rho \times p\times V_{1}}{M\times V_{2}}\times 1000$$
where *C* (ppm) stands for the target gas concentration, *Ø* stands for the required gas volume fraction, *ρ* (g mL^−1^) stands for the density of the liquid, *p* stands for the purity of the liquid, *V*_1_ (μL) stands for the volume of liquid, *V*_2_ (L) stands for the volume of the chamber, and *M* (g mol^−1^) stands for the molecular weight of the liquid.

The change in resistance of the sensor was measured with a multimeter. The response of the sensor was defined as follows:(2)$$S=\frac{R_{g}-R_{a}}{R_{a}}\times 100\%$$
where *R_a_* is the resistance of the sensor in the air, and *R_g_* is the resistance of the sensor after it passes into the target gas.

## 3. Results

### 3.1. Characterization of ZnO/CBC

The XRD patterns of ZnO/CBC and CBC with different ZnO precursor contents are shown in Figure 2. ZnO/CBC showed sharp diffraction peaks at 2θ = 31.8°, 34.4°, 36.3°, 47.5°, 56.6°, 62.9°, 66.4°, 67.9°, 69.1°, 72.6°, and 77.0°, which were by the (100), (002), (101), (102), (110), (103), (200), (112), (201), (004), and (202) crystal planes, respectively. The strongest peak occurred on the (101) crystal plane. All diffraction peaks correspond to the standard hexagonal wurtzite structure (JCPDS-99-0111) [31,32]. No diffraction peaks from other phases or impurities were observed. These results indicated that pure ZnO structures were formed through precipitation. Since the amorphous carbon was prepared and its crystallinity is low, wide peaks with low signal strength appeared near 22° and 42° [33,34,35]. The diffraction peak strength was weaker than ZnO, but this could not be shown in the XRD pattern of ZnO/CBC.

Figure 3 shows the SEM images of ZnO/CBC and pure carbon with different ZnO precursor contents. It can be seen from Figure 3b–e that ZnO has been successfully loaded on CBC. ZnO exhibited a flower-like structure when the precursor content was small. With the ZnO precursor content increasing, ZnO gradually changed into a rod-like structure; meanwhile, the aspect ratio also became larger. This was due to the increase in ammonia solution, which promoted oxidation. Zinc preferentially grew on the C axis [36]. Figure 3a shows that the surface of CBC was relatively smooth after high-temperature carbonization. However, after loading ZnO, the surface of CBC becomes rough. EDS was used to conduct an elemental analysis of ZnO/CBC-60%. It can be seen from Appendix A that C, O, and Zn have been evenly distributed on ZnO/CBC-60%, and the atomic ratios of C, O, and Zn were 77.50%, 11.84%, and 10.66%, respectively. Since aluminum foil was used as the substrate, the Al signal appeared.

In order to further understand the structure of ZnO/CBC-60%, TEM characterization was performed. As shown in Figure 4a,b, it was found that there were different lattice fringes at the boundary of CBC and ZnO. The lattice fringes near 0.136 and 0.282 nm correspond to the (201) and (100) crystal faces of ZnO, respectively [16,37]. The lattice fringes near 0.202 and 0.350 nm correspond to the (101) and (002) crystal faces of graphite, respectively [38]. The lattice fringes of ZnO and CBC were interwoven, which also confirmed the construction of heterojunctions between ZnO and CBC.

As shown in the Raman spectrum in Figure 5a, it can be observed that peak D and peak G of typical carbon materials were located at 1340 cm^−1^ and 1584 cm^−1^, respectively, where peak D represents the sp^3^ hybrid carbon of disordered or defective carbon, while peak G corresponds to the sp^2^ hybrid carbon of the graphite structure [39]. The relative strength ratio (I_D_/I_G_) of peak D and peak G indicates the degree of graphitization of the material. The I_D_/I_G_ values of pure carbon and ZnO/CBC-60% were 0.90 and 0.79, respectively, which were much higher than the I_D_/I_G_ values of typical graphite, indicating that the prepared carbon materials were not highly crystalline and disordered materials exist. This was consistent with the observation with XRD. Moreover, the addition of ZnO improved the degree of disorder and defects in the structure of carbon materials [40]. The peaks at 80 cm^−1^ and 427 cm^−1^ in ZnO/CBC-60% correspond to $E_{2}^{low}$ and $E_{2}^{high}$, respectively. $E_{2}^{low}$ was affected by the Zn^2+^ gap defect, and $E_{2}^{high}$ was affected by O^2−^ vacancy [41]. The simultaneous appearance of characteristic peaks of ZnO and carbon materials confirmed the successful preparation of ZnO/CBC. At the same time, it can be observed from Figure 6b that the D peaks and G peaks of ZnO/CBC-60% had a certain wave number displacement relative to pure carbon materials. It can be concluded that the heterojunction constructed by ZnO and cellulose-based carbon induces charge transfer between them [22].

In order to verify the elemental composition and surface chemical information of ZnO/CBC-60%, the materials were characterized with XPS. Figure 6a shows that ZnO/CBC-60% was mainly composed of Zn, C, and O elements. Figure 6b was the high-resolution spectrum of Zn^2p^. In the figure, the peaks of Zn^2p1/2^ and Zn^2p3/2^ were 1044.0 eV and 1021.1 eV, respectively, and the binding energy distance before the two peaks was 22.9 eV, which proved the existence of divalent zinc ions [42]. Figure 6c shows the high-resolution spectrum of O 1 s. O 1 s can be deconvolved into three peaks centered on 529.5 eV, 531.2 eV, and 532.2 eV. The peak at 529.5 eV is attributed to the O^2−^ ion in the Zn–O bond in the ZnO wurtzite structure, and the peak at 531.2 eV is attributed to the Zn–O–C bond and the adsorbed hydroxyl group or water. The peak at 532.2 eV is attributed to the carbonate (C–O/C=O) species [43]. As shown in Figure 6d, three peaks appeared at 283.6 eV, 285.7 eV, and 288.3 eV which belonged to the Zn-C bond, Zn–O–C bond, and C=O bond, respectively, indicating that the performance of ZnO/CBC-60% was different from that of pure ZnO [44].

Mott–Schottky curves were used to determine the semiconductor type of the material to better explain the sensing mechanism of the material. Appendix A shows the Mott–Schottky curves of CBC and ZnO/CBC-60% at different frequencies measured at room temperature. It can be seen from the figure that the slopes of all curves are negative, indicating that CBC was a p-type semiconductor, and the prepared ZnO/CBC-60% also exhibited p-type semiconductor properties.

### 3.2. Results of Sensing Tests

Firstly, the gas sensing response of ZnO/CBC with different contents of CBC and ZnO precursors to 200 ppm ammonia at room temperature and 60% relative humidity were tested. When the sensor was in an air environment, the resistance of the sensor was at a stable value (*R_a_*). When exposed to ammonia, the resistance of the sensor increased and reached a maximum value (*R_g_*). When the sensor was exposed to air again, the resistance of the sensor gradually returned to the initial value (*R_a_*). The resistance of CBC and ZnO/CBC increased gradually in the ammonia environment and decreased gradually in air, which shows p-type semiconductor behavior. After testing the sensor, Figure 7a shows that when the ZnO precursor content was 60%, the sensor’s gas sensing response to 200 ppm ammonia was the highest, reaching 27%. Therefore, zinc acetate/cellulose (wt%) = 150 wt% was selected as the optimal ratio for preparing the required sensing material. The raw CBC also showed a certain gas sensing response to an ammonia concentration of 200 ppm. After the high-temperature carbonization of microcrystalline cellulose, the CBC is a p-type semiconductor, which also has a good effect on ammonia sensing. This result is consistent with previously reported [39] biomass-based carbon sensing of ammonia. ZnO and CBC formed a p-n heterojunction after compounding, in which CBC plays a dominant role.

In addition, the ZnO/CBC-60% sensor was placed in the environment of methanol, isopropyl alcohol, ethanol, and formaldehyde at 200 ppm at room temperature to test the gas sensing performance of the sensor to these gases, to verify the selectivity of the sensor to ammonia. It can be seen from Figure 7b that the gas sensing response of the sensor to ammonia is dozens of times that of other gases, indicating that the prepared ZnO/CBC-60% sensor has good selectivity to ammonia. This may be attributed to the fact that the p-n heterostructure constructed between ZnO and CBC provided more active sites for the selective adsorption of ammonia molecules. Good selectivity improves the ZnO/CB-60% sensor’s anti-interference ability, allowing it to accurately and quickly detect ammonia concentration in an environment where multiple gases exist.

In order to further test the gas sensing performance of ZnO/CBC-60% for ammonia, the sensor was put in the ammonia environment with five concentrations of 25 ppm, 50 ppm, 100 ppm, 150 ppm, and 200 ppm at room temperature for the gas sensing test. The gas sensing response of the sensor to ammonia with a concentration of 25–200 ppm at room temperature is shown in Figure 8a. As the ammonia concentration increased, the gas sensing response of the sensor increased, which was in line with the expected goals of this experiment. Stability is one of the key parameters of gas sensing. We tested the stability of the ZnO/CBC-60% sensor by placing the sensor in a 200 ppm ammonia environment for five rounds of gas sensing tests. As shown in Figure 8b, after five sensing cycles, the gas sensing performance of the ZnO/CBC-60% sensor had not weakened, indicating that the sensor had good repeatability.

The relative humidity is an important factor affecting the performance of the gas sensor [45]. As shown in Figure 9a, with the increase in relative humidity, the gas sensitivity of the ZnO/CBC-60% sensor to ammonia was correspondingly weakened, which may be due to the fact that H_2_O molecules in the test environment preempted the active sites so that NH_3_ molecules could not fully react with the materials, resulting in the decrease in the gas sensing response [23].

Stability is one of the important indicators of a gas sensor, which reflects whether the gas sensing capability of the gas sensor can maintain stability over a period of time to extend the service life of the equipment. The ZnO/CBC-60% sensor was put in an ammonia concentration environment of 200 ppm for one week. As can be seen from Figure 9b, the gas sensing response of the sensor to ammonia had not changed greatly, which indicated that the sensor had good stability.

The sensing performance of the as-developed ZnO/CBC and most of the previously reported ZnO-, WS_2_-, or TiO_2_-based ammonia gas sensors are summarized in Table 1. Compared with previous ammonia sensors, the ammonia sensor constructed with ZnO and CBC can detect ammonia at room temperature. At the same time, the gas sensing response has been enhanced to a certain extent, which can mainly be ascribed to the unique chemical properties of the two substances and the formation of a p-n heterojunction. However, the response and recovery ability should be further enhanced.

## 4. Discussion

In order to prove that the gas sensing response of ZnO/CBC-60% was the result of the reaction with NH_3_ molecules, the material was analyzed with in situ DRIFT spectroscopy. As shown in Figure 10a, after introducing the mixed gas of NH_3_ and N_2_, it can be observed that adsorption peaks of coordinated NH_3_ species corresponding to Lewis sites appeared near 929, 964, 1619, and 3334 cm^−1^ [49,50,51], and these peaks gradually became stronger with the increase in adsorption time. Figure 10b shows that the peak value was obviously weakened after N_2_ purging. These phenomena show that the adsorption and desorption of ammonia occurred on the surface of ZnO/CBC-60%, which explained the gas sensing response of ZnO/CBC-60% to ammonia.

In the existing literature, the surface charge control model is often used to explain the sensing mechanism of resistive gas sensors. When the ZnO/CBC-60% sensor was exposed to air, oxygen molecules in the air stuck to the surface of the material and trapped electrons to generate $O_{2}^{-}$, which was revealed by Formulas (3) and (4). During this process, a depletion layer was formed on the surface of the ZnO/CBC-60%. At the same time, it can be seen from the Mott–Schottky curve that CBC is a p-type semiconductor, while ZnO is a typical n-type semiconductor. As illuminated in Figure 11, the combination of the two creates a p-n heterojunction. Due to the difference in the concentration of electrons and holes in the two materials, the free electrons from the ZnO conduction band will diffuse to CBC, and the holes in CBC will diffuse to ZnO until the Fermi level reaches a new equilibrium [52,53]. This process makes the energy band bend, forming a narrow depletion layer, an electron accumulation layer on the CBC side, and a hole accumulation layer on the ZnO side. As shown in Formulas (5) and (6), when the ZnO/CBC sensor was exposed to ammonia, ammonia molecules reacted with adsorbed oxygen, releasing electrons that neutralize holes and further expand the depletion layer on the CBC side, leading to a significant increase [54] in the resistance of the sensor, which was consistent with the phenomenon in the gas sensing test.
(3)$$O_{2}\left(in\;air\right)\;\to \;O_{2}\left(ads\right)$$
(4)$$O_{2}\left(ads\right)+e^{-}\;\to \;O_{2}^{-}\left(ads\right)$$
(5)$${NH}_{3}\left(in\;air\right)\;\to \;{NH}_{3}\left(ads\right)$$
(6)$${4NH}_{3}\left(ads\right)+{3O}_{2}^{-}\;\to \;{2N}_{2}+{6H}_{2}O+{3e}^{-}$$

## 5. Conclusions

In this experiment, ZnO was loaded on CBC, and the gas sensing response of ZnO/CBC loaded with different ZnO contents to ammonia at room temperature was studied with various contents of ZnO precursors. The composites were characterized with SEM, TEM, XRD, Raman, and XPS. The results showed that ZnO had been successfully loaded on CBC. Subsequently, the gas sensing performance showed that the ZnO/CBC with 60% ZnO precursor content had a better gas sensing response to 200 ppm ammonia at room temperature, reaching 27%. In addition, through five consecutive rounds of gas sensing response tests in a 200 ppm ammonia environment, it was found that ZnO/CBC-60% exhibited good stability. At the same time, by comparing the gas sensing performance of several VOC gases, it was found that the material had good selectivity for ammonia. In situ DRIFT spectroscopy proved that the sensing material did react with ammonia, and the ZnO/CBC-60% could provide more active sites for ammonia molecules because of the construction of a heterojunction between ZnO and CBC. Therefore, compared with previous ammonia sensors, ZnO/CBC ammonia sensors have improved the gas sensing performance to a certain extent and overcome the problem that pure ZnO ammonia sensors cannot sense ammonia at room temperature, reducing energy consumption. This study provides a new idea for the design and synthesis of biomass carbon–metal oxide composites.

## Figures and Tables

**Figure 1 nanomaterials-13-03151-f001:**
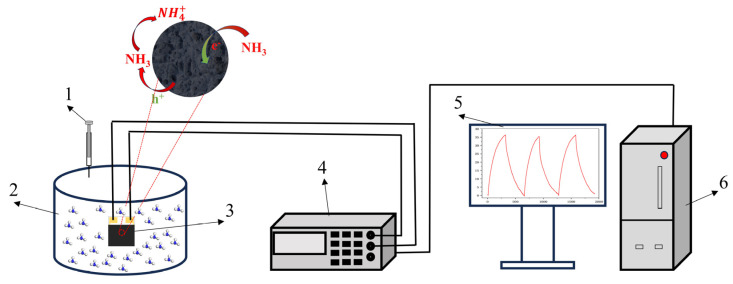
Schematic diagram of gas detection. (1) Syringe, (2) reaction chamber, (3) interdigital electrode, (4) Victor 8145C TRMS digit multimeter, (5) display screen, and (6) mainframe computer.

**Figure 2 nanomaterials-13-03151-f002:**
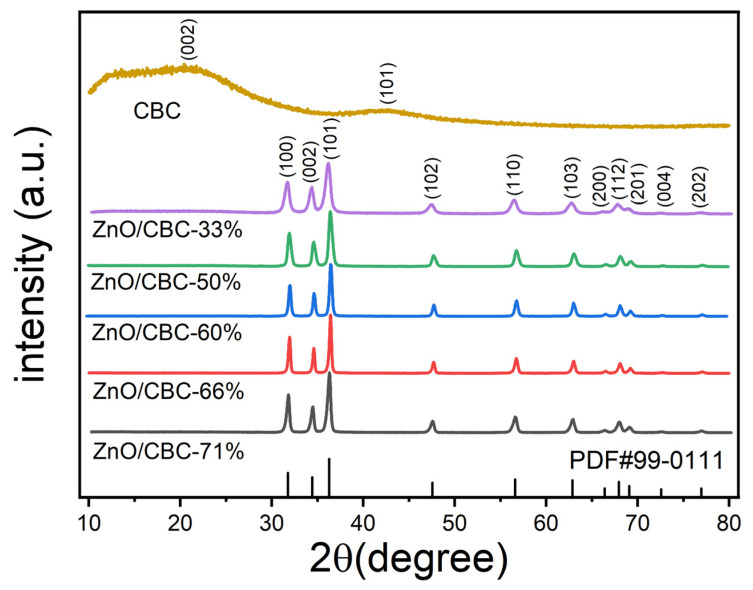
XRD patterns of ZnO/CBC with different ZnO contents and CBC.

**Figure 3 nanomaterials-13-03151-f003:**
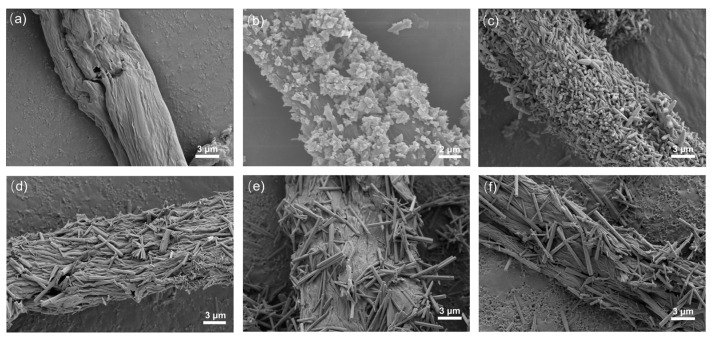
SEM image of (**a**) CBC, (**b**) ZnO/CBC-33%, (**c**) ZnO/CBC-50%, (**d**) ZnO/CBC-60%, (**e**) ZnO/CBC-66%, and (**f**) ZnO/CBC-71%.

**Figure 4 nanomaterials-13-03151-f004:**
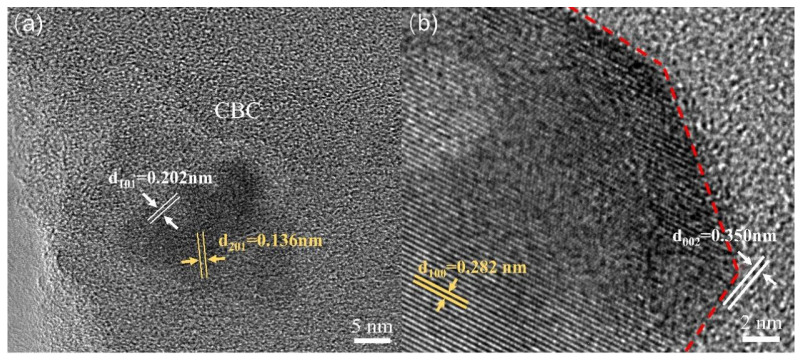
TEM images of ZnO/CBC-60%. (**a**) The (201) and (100) crystal faces of ZnO, (**b**) The (101) and (002) crystal faces of graphite.

**Figure 5 nanomaterials-13-03151-f005:**
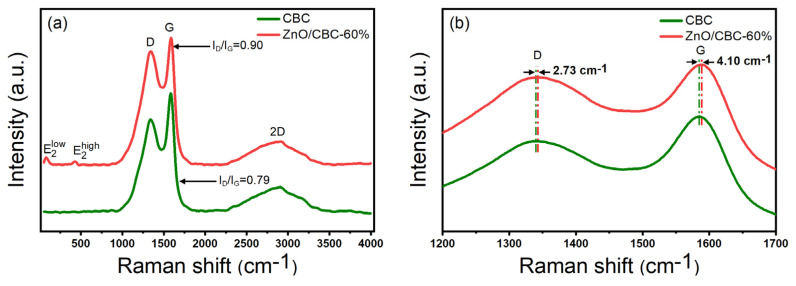
(**a**) Raman spectra of ZnO/CBC-60% and CBC; (**b**) partial enlarged Raman spectra of ZnO/CBC-60% and CBC.

**Figure 6 nanomaterials-13-03151-f006:**
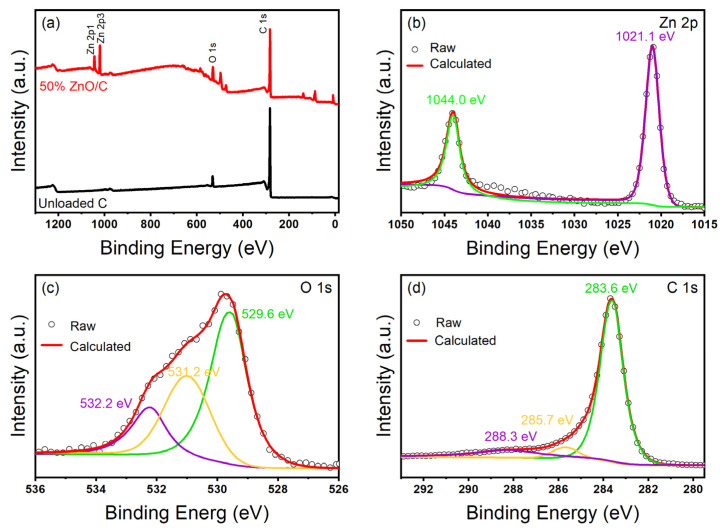
XPS patterns of ZnO/C-60% and CBC. (**a**) Full spectrum of sample, (**b**) Zn 2p XPS spectra of ZnO/CBC-60%, (**c**) C 1s XPS spectra of ZnO/CBC-60%, and (**d**) O 1s XPS spectra of ZnO/CBC-60%.

**Figure 7 nanomaterials-13-03151-f007:**
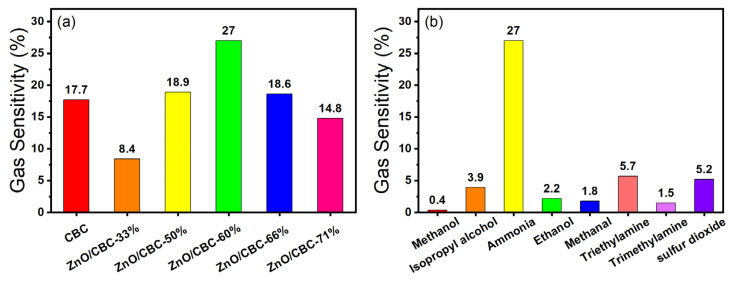
(**a**) Response of different amounts of zinc oxide and CBC to ammonia at 200 ppm concentration; (**b**) selectivity of ZnO/CBC-60% for ammonia.

**Figure 8 nanomaterials-13-03151-f008:**
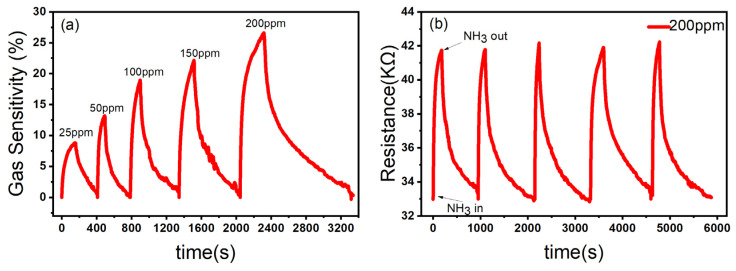
(**a**) Response of ZnO/CBC-60% to ammonia with concentration of 25–200 ppm; (**b**) repeatability evaluation of ZnO/CBC-60% for 200 ppm ammonia at RT.

**Figure 9 nanomaterials-13-03151-f009:**
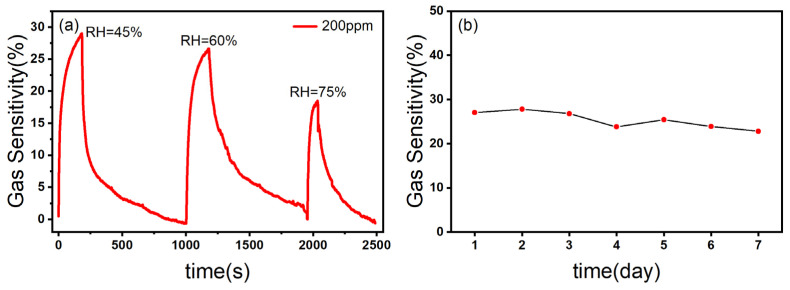
(**a**) The response curves of ZnO/CBC-60% towards 200 ppm ammonia at different RHs; (**b**) stability test of ZnO/CBC-60% for 200 ppm ammonia within one week at RT.

**Figure 10 nanomaterials-13-03151-f010:**
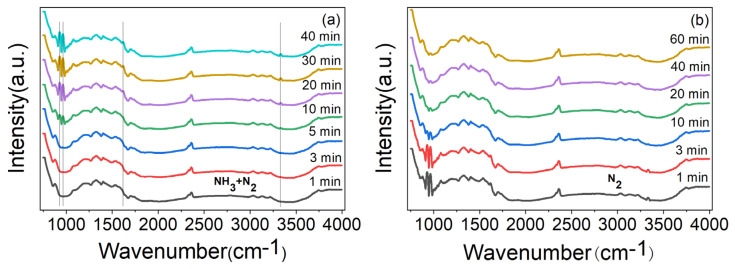
(**a**) In situ DRIFT spectroscopy of ZnO/CBC-60% by passing NH_3_ + N_2_ mixture gases at RT for 40 min and (**b**) purged by N_2_ at RT.

**Figure 11 nanomaterials-13-03151-f011:**
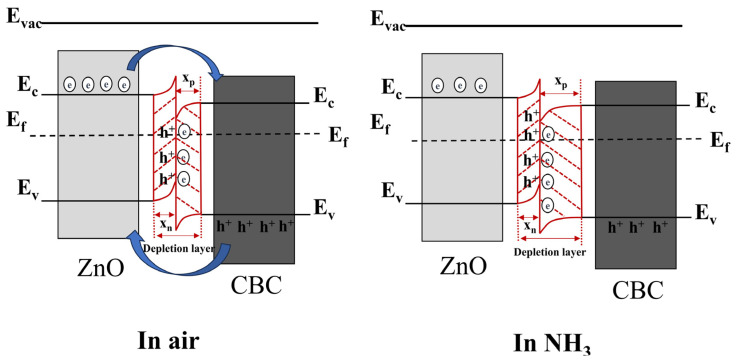
The energy band structure diagram of n-type ZnO/p-type CBC hetero-contact. The diffusion process of free electrons and holes is represented by blue arrows. The formation of heterojunctions is represented by red lines.

**Table 1 nanomaterials-13-03151-t001:** Comparison of the sensing performance of various gas sensors toward NH_3_.

Material	Conc.	Res. ^a^	Res./Rec. Time (s)	Temp. (°C)	Ref.
TiO_2_/Ti_3_C_2_T_x_	10 ppm	3.1%	33/277	RT	[5]
WS_2_	250 ppm	2.5%	200/412	RT	[46]
ZnO	3 ppm	44%	15/14	250	[47]
ZnO/rGO	10 ppm	1.2%	78/84	RT	[48]
Mn-ZnO	20 ppm	7%	4/10	150	[15]
Y-doped ZnO	150 ppm	66%	58/87	250	[16]
ZnO/CBC	25 ppm	9.2%	154/254	RT	This work

^a^ Response defined as (Rgas − Rair)/Rair × 100%.

## Data Availability

The data presented in this study are available on request from the corresponding author.

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
