# Peer review of "Zinc Oxide-Loaded Cellulose-Based Carbon Gas Sensor for Selective Detection of Ammonia"

_nanomaterials, 2023, doi:10.3390/nano13243151_

Round 1
Reviewer 1 Report
Comments and Suggestions for Authors
Before the paper could be published, the authors should address the following questions and notes:
1) Why did the authors choose ZnO to make a sensor for ammonia? Are there other chemical compounds serving for this purpose? What data can be found in literature on sensitivity and selectivity of the NH3 sensors, to compare with the material invented by the authors? It must be reflected in the Introduction and in Conclusions.
2) The experiment should be seriously rewritten in order to make a style uniform and clear. E.g., in the present form it is difficult to understand how to make materials with different ZnO tot CBC ratios. Also, did authors check the reproducibility of their method?
3) It is important to address the chemical nature of the interaction of ammonia with ZnO. Which chemical processes can proceed?
4) Why did the authors choose several chemical compounds to check the selectivity towards NH3? What is known for the sensor’s response to other nitrogen-containing compounds like aliphatic and aromatic amines, various nitrogen and sulfur oxides?
5) How can the authors explain the data presented on Fig. 7a? The sensitivity of CBC lays between the sensitivity of different ZnO/CBC composites and the difference between them is not very significant.
6) Authors should carefully check and improve English, there are too many grammatical mistakes.
Comments on the Quality of English LanguageAuthors should carefully check and improve English, there are too many grammatical mistakes.
Author Response
Response to Reviewer 1 Comments
- Summary Thank you very much for taking the time to review this manuscript. Please find the detailed responses below and the corresponding revisions/corrections highlighted/in track changes in the re-submitted files.
- Questions for General Evaluation Reviewer’s Evaluation Response and Revisions Does the introduction provide sufficient background and include all relevant references? Must be improved Are all the cited references relevant to the research? Can be improved Is the research design appropriate? Can be improved Are the methods adequately described? Must be improved Are the results clearly presented? Must be improved Are the conclusions supported by the results? Can be improved
- Point-by-point response to Comments and Suggestions for Authors Comments 1: [Why did the authors choose ZnO to make a sensor for ammonia? Are there other chemical compounds serving for this purpose? What data can be found in literature on sensitivity and selectivity of the NH3 sensors, to compare with the material invented by the authors? It must be reflected in the Introduction and in Conclusions.] Response 1: Thank you for pointing this out. We agree with this comment. Therefore, we have added some content to the introduction and conclusion. Introduction (Page 1, Paragraph 2, Line 1-8). Conclusion (Page 10, Paragraph 1, Line 13-16). We also added a table 1 to compare with previously reported ammonia sensors Comments 2: [The experiment should be seriously rewritten in order to make a style uniform and clear. E.g., in the present form it is difficult to understand how to make materials with different ZnO tot CBC ratios. Also, did authors check the reproducibility of their method?] Response 2: Agree. We have rewritten experimental steps in Page 2, Paragraph 4, Line 1-11. We checked the reproducibility of their method, it is feasible. Comments 3: [It is important to address the chemical nature of the interaction of ammonia with ZnO. Which chemical processes can proceed?] Response 3: Thank you for pointing this out. We agree with this comment. Therefore, we have added some chemical processes in Page 9, Paragraph 2, Line 1-5, Formulas (1)-(4). Comments 4: [Why did the authors choose several chemical compounds to check the selectivity towards NH3? What is known for the sensor’s response to other nitrogen-containing compounds like aliphatic and aromatic amines, various nitrogen and sulfur oxides?] Response 4: Thank you for pointing this out. we tested the gas sensing performance of triethylamine, trimethylamine and sulfur dioxide at a concentration of 200 ppm respectively. The results are shown in the Fig. 7. Compared with ammonia, the gas sensing response of the sensor to these three gases is not good. Comments 5: [How can the authors explain the data presented on Fig. 7a? The sensitivity of CBC lays between the sensitivity of different ZnO/CBC composites and the difference between them is not very significant.] Response 5: Thank you for pointing this out. we read some literature and came to the following conclusion in Page 7, Paragraph 3, Line 12-17. 4. Response to Comments on the Quality of English Language Point 1: Authors should carefully check and improve English, there are too many grammatical mistakes. Response 1: Thank you for pointing this out. We have carefully checked and fixed grammatical errors. (in red)

Reviewer 2 Report
Comments and Suggestions for Authors
This manuscript deals with the development of NH3 sensor based on carbon-supported Zn oxide. There is a sound characterisation of the active sites, although novelty is somewhere limited. However, I consider this contribution quite relevant, for two main factors. (i) the combined and synergistic application of ZnO on one hand, and carbon-based technological materials may provide new and interesting applications, non possible without a strong interactions between the two materials; and (ii) the green approach applied by the authors, in using green cellulosa as starting precursor of carbon materials. I would even add a suggestion to the author, in that raw or waste materials may as well act a precursors of hard carbons, with even less impact on earth’s resources.
The main point I appreciate in this manuscript is the thorough and complete characterisation of the final materials, using a multi technique approach. And this may be considered one of the most important contribution to the field. In this respect, I evaluated the appropriateness of references reported by the authors.
I have no additional comments to tables and figures, which are pretty appropriate. The only comment I think it has to to be answered by the authors refers to XPS spectra deconvolution. XPS data are, to my opinion, of primary importance in the research body; however, the authors should report goodness of fit, and show differences between raw and calculated data. Apart this point, the evidences and arguments presented are quite solid, and the only main question to be addressed and answered if the former on XPS data.
The manuscript proposed by Mr.Xi-Ping Luo and coworkers is to be considered for publication on a prestigious journal as Nanomaterials, provided my comment is fully addressed.
Author Response
Response to Reviewer 2 Comments
- Summary Thank you very much for taking the time to review this manuscript. Please find the detailed responses below and the corresponding revisions/corrections highlighted/in track changes in the re-submitted files.
- Questions for General Evaluation Reviewer’s Evaluation Response and Revisions Does the introduction provide sufficient background and include all relevant references? Yes Are all the cited references relevant to the research? Can be improved Is the research design appropriate? Yes Are the methods adequately described? Yes Are the results clearly presented? Can be improved Are the conclusions supported by the results? Can be improved
- Point-by-point response to Comments and Suggestions for Authors Comments 1: [A The only comment I think it hasto to be answered by the authors refers to XPS spectra deconvolution. XPS data are, to my opinion, of primary importance in the research body;however, the authors should report goodness of fit, and show differencesbetween raw and calculated data. Apart this point, the evidences andarguments presented are quite solid, and the only main question to be addressedand answered if the former on XPS data.] Response 1: Thank you for pointing this out. we have redrew Fig. 6 to report goodness of fit, and show differences between raw and calculated data.
